# Effects and Mechanism of Hyperbranched Phosphate Polycarboxylate Superplasticizers on Reducing Viscosity of Cement Paste

**DOI:** 10.3390/ma17081896

**Published:** 2024-04-19

**Authors:** Jing Chen, Changhui Yang, Yan He, Futao Wang, Chao Zeng

**Affiliations:** 1College of Materials Science and Engineering, Chongqing University, Chongqing 400030, China; chenjing0800@hotmail.com (J.C.); wangfutao619@163.com (F.W.); whutzengchao@163.com (C.Z.); 2School of Civil Engineering, Suzhou University of Science and Technology, Suzhou 215011, China

**Keywords:** hyperbranched phosphate PCEs, cement, adsorption behavior, viscosity reducing

## Abstract

The adsorption behavior and dispersing capability of hyperbranched phosphated polycarboxylate superplasticizers (PCEs) containing phosphate monoester and phosphate diester were investigated. The hyperbranched structures were constructed using a special monomer dimethylaminoethyl methacrylate (DMAMEA) to create the branches during the polymerization. Meanwhile, the polymer architectures were tailored by varying the content of phosphate monoester and phosphate diester in the backbone via free radical solution polymerization. In contrast to comb-like PCE, hyperbranched PCEs presented a weaker dispersion capability at w/c = 0.29, but with a lower water-to-cement ratio (w/c), the hyperbranched PCEs exhibited a better dispersion capability than the comb-like PCEs. The dynamic light scattering (DLS) and transmission electron microscope (TEM) analysis showed that the adsorption layer of hyperbranched PCEs were thicker than that of comb-like PCEs. A thicker adsorption layer thickness generated thinner diffusion water layer thickness. The increase of the free water amount due to the thinner water diffusion layer is the key mechanism for improving the dispersibility and decreasing the viscosity of cement paste.

## 1. Introduction

Recently, PCEs have been adopted into concrete in the construction industry because of their great water reduction behavior and molecular designable performance [1,2,3,4,5]. The low viscosity and high fluidity are the basic characteristics of modern concrete; these bring great demands for PCE, cementing material, and coarse and fine aggregate [2,6,7]. These optimized concrete mixes tend to be more “sticky” and have poor flowability, which negatively influences concrete mixing [7,8,9,10].

The stickiness of modern high-strength concrete may be related to the energy dissipation [11,12]. The macroscopic viscosity is highly linked to the increase in the local viscosity [13,14,15,16,17]. The mineral and chemical admixtures that are used to decrease the viscosity of cement paste can be ascribed to the theory of the closest packing and spatial structure of PCEs, as it influenced the water film thickness and viscosity of interstitial fluid [18,19,20]. The hyperbranched PCEs possess a novel branched structure that exhibits a lower viscosity [11,21]. Amin [22] found that the hyperbranched polyesteramides reduced the consistency and increased the compressive strength of the cement pastes. Liu and Navarro-Blasco [23,24] reported that the star-shaped PCE exhibited better fluidity, fluidity retention, and water reduction. Huang [11] synthesized a viscosity-reducing type of hyperbranched PCE with dimethylaminoethyl methacrylate [25,26]. The phosphate PCE showed a higher Ca binding capacity [27,28,29]. 

The relationship between the adsorption layer thickness and water film thickness has rarely been investigated in the literature [30,31,32,33]. For the phosphate PCE, the phosphate diester has often been ignored, including the fact that the phosphate diester leads to the formation of a small amount of crosslinked copolymer [27,28]. In this work, the hyperbranched phosphate PCE was synthesized by substituting carboxylate with phosphate ester groups. Additionally, the phosphate PCEs were prepared by controlling the phosphate monoester equality and phosphate diester equality. The adsorption layer thickness and viscosity-reducing properties of synthesized PCEs were studied, and the rheology of fresh cement paste was also investigated.

## 2. Materials and Methods

### 2.1. Material

Industrial-grade isobutyl alcohol polyoxyethylene ether (IPEG) with a molecular weight of 2400 (average polymerization degree: ~54), and 2-(acryloyloxy) ethyl phosphate (MOEP) and 5-hydroxypentyl prop-2-enoate phosphate (HPEP), were manufactured by Liaoning Kelong Fine Chemical Company (Liaoyang, Liaoning Province, China), and Guangzhou Kinde Chemical Materials Company (Guangzhou, Guangdong Province, China), respectively. Analytical grade acrylic acid (AA), 2-(Dimethylamino)ethyl methacrylate (DMAEMA), Maleic anhydride (MA), N,N′-Methylenebisacrylamide (MBA), potassium persulfate (KPS), mercapto acetic acid (TGA), CaSO_4_·2H_2_O, KOH, K_2_SO4, Na_2_SO_4_, and ascorbic acid (VC) were from Sinopharm Chemical Company (Shanghai, China). The schematic drawings of the unsaturated phosphate monomers are exhibited in Figure 1. The phosphate monoester, phosphate diester, and free phosphoric acid of unsaturated phosphate ester are shown in Table 1. 

Ordinary Portland cement classified as P Ⅱ 52.5 was purchased from Sichuan E-Sheng Cement Group Co., Ltd. Chemical (Emei, Sichuan Province, China) compositions of cement are tabulated in Table 2.

### 2.2. Synthesis of PCEs

(1)Synthesis of hyperbranched PCEs

The amounts of unsaturated phosphate and TGA were shown in Table 3. IPEG (202.5 g), MA (3.70 g), and deionized water (204.00 g) were added into a four-necked flask and the temperature was gradually raised to 45 °C. AA (18.98 g). DMAEMA (1.27 g) and TGA were dissolved in deionized water (30.00 g). APS (1.94 g) was dissolved in deionized water (28.00 g) to obtain oxidant solution. The monomer solution and oxidant solution were dropped into the reaction flask for 2 h and 2.5 h. 

(2)Synthesis of comb-like PCE

The conventional comb-like PCE (P1) was synthesized by monomer solution (which consisted of the recipe of P2 without adding DMAEMA), oxidant solution, and reductant solution (which consisted of 0.35 g of ascorbic acid and 30.00 g of deionized water). 

### 2.3. Characterization of PCEs

The PCEs were purified by dialysis via a 3500 Da cellulose membrane. 

FT-IR spectra were measured by a Nicolet Avatar 370 spectrometer. The molecular weight, PDI, intrinsic viscosity ([η]), and hydrodynamic radius were determined by GPC. The hydrodynamic radius was determined by dynamic light scattering. The crystallizing point (Tc) and melting point (Tm) were carried out using a DSC. The phase transition temperatures were recorded when the values of the DSC curves were maximized. Surface tension was tested via an OCA40 Micro surface contact angle tester.

### 2.4. Dispersion of PCEs

Cement paste was mixed at w/c of 0.29 at 20 °C. The fluidity was determined via a mini slump experiment. The Marsh time of the cement paste was measured through the funnel. The rheological parameters were measured via a rheometer. 

### 2.5. Adsorption Layer Thickness

The adsorption layer thickness (ALT) was measured by simulated cement dispersion. The surface of nano-silica was modified with Ca^2+^ [11]. The particle sizes of silica-CSH before and after the adsorption of PCEs were tested by dynamic light scattering (DLS, ALV/CGS-3, Langen, Germany).

### 2.6. Apparent Adsorption Amount

The fresh cement paste was centrifuged and filtered. The supernatant solution was diluted for total organic carbon analyzer [34,35].

### 2.7. Viscosity of Aqueous Solution and Simulated Pore Solution

Viscosity was also tested via a rotational rheometer. The concentration was selected based on the viscosity-reducing PCEs’ accounts for 30% to 60% of PCE.

## 3. Results

### 3.1. Structure Characterization of PCEs

Figure 2 exhibits FT-IR spectra of comb-like and hyperbranched PCEs. The new adsorption bands at 1020 cm^−1^ are ascribed to the C-N group of the DMAEMA units. The new peak at 1282 cm^−1^ is assigned to the P=O deformation vibration. PM-4 and PD-3 had a stronger absorption peak than others at 842 cm^−1^ due to the HPEP over three methylene than MOEP. 

The molecular weight, intrinsic viscosity, Mark–Houwink α value, crystallizing point (Tc), and melting point (Tm) of PCEs were listed in Table 3. As shown in Table 3, the number-average molecular weight of PCEs were relatively close. In contrast to P1, the Tc and Tm of modified PCEs increased distinctly, and the intrinsic viscosity reduced distinctly. The hyperbranched PCEs with the branched structure were clearly revealed by Mark–Houwink α value due to the different Mark–Houwink α value of hyperbranched PCEs and comb-like PCEs. For the ordinary branched polymer, Mark–Houwink α usually decreased to about 0.5, while 0.21–0.44 meant a hyperbranched structure [25,26]. Table 3 shows the Mark–Houwink α value decreased with phosphate diester increased for PM series PCEs except PM-4, which indicated that the branching degree of hyperbranched structure increased while Mark–Houwink α value decreased. It can be explained that the cross-linked structure formed by the phosphate diester led to the main chain being shortened and the side chain stretched, but the further increase of the phosphate diester reached the branching limit due to PM-4 being prone to intermolecular hydrophobic associations to form a mixed aggregate. Meanwhile, the intrinsic viscosity of PM series PCEs had the same changing trend as Mark–Houwink α value. This revealed that the branching degree of the hyperbranched structure and intrinsic viscosity increased and decreased with the decrease of phosphate monoester amount, respectively.

### 3.2. Surface Tension

The surface tension of the PCEs’ solution is shown in Figure 3. In reality, the surface tension of the PCEs’ solution correspondingly decreased with the concentration of the increased PCEs’ solution. The surface tension of the PCEs followed the order of PM-4 < PM-3 < PM-2 < PD-3 < PD-2 < PM-1 < PD-1 < P2 < PB-1 < P1. For the surface tension of the PM series of PCEs, with the increases of the phosphate diester, the surface tension gradually decreased, indicating that the cross-linked structure cladded the hydrophilic groups. PM-4 had stronger hydrophobicity than others due to the HPEP having over three methylene more than MOEP. The effect of the phosphate monoester on surface tension was also investigated. It can be concluded that the decrease of the phosphate monoester induced a decrease in surface tension due to the declined anchor groups, the weakened steric repulsive force, and the increased intermolecular and intramolecular molecules aggregation tendency. This conclusion further supported the relationship between the branching degree and intramolecular and intermolecular forces.

The surface tension of PB-1 is between those of P1 and P2, which implies that the PB-1 possessing the hydrogen bonds between H, N, and O atoms in chains are easily formed to increase the intermolecular interaction and aggregation, thus leading to an increase of surface tension. However, the lower surface tension of hyperbranched possession can reduce the interfacial energy and easily form bubbles in fresh concrete, which might be helpful to dispersing cement particles.

### 3.3. Fluidity 

Figure 4 presents the effect of the hyperbranched PCEs on the dispersing ability. Figure 4a shows that the hyperbranched PCEs require higher dosage than comb-like PCEs (P1) at the same fluidity for w/c = 0.29, which indicates that the hyperbranched structure had negative effects on the dispersing ability of PCEs. However, with the increases of the phosphate diester, the dispersing ability gradually increased for PM series PCEs except PM-4, implying that the higher the degree of branching in the hyperbranched structure, the stronger the dispersing ability. The dosage of PM-4 was higher than others due to the fact that PM-4 had lower surface tension and lower gas–liquid interfacial energy. Therefore, the cement pastes presented a good fluidity at a higher dosage of PM-4. The results of the phosphate monoester on the effect of dispersing ability, which indicated that the dispersing ability increased, reached a plateau with a further increasing phosphate monoester. This phenomenon further confirmed the phosphate anchor group was advantageous for the performance of PCE in cement paste such as the dispersibility and water reducibility. It is interesting that the dosage of PB-1 is close to PM-2, which suggests that the higher the branching degree of the hyperbranched structure, the larger the steric hindrance and the stronger the dispersing ability. In other words, if the PCE is imbedded with a hyperbranched structure, the charge density of the PCE will be decreased at the same dispersing ability.

As can be seen from Figure 4b, the hyperbranched PCEs had lower dosage required than comb-like PCEs at w/c = 0.20, and the dispersing ability of PCEs increased with the branching degree of the increased hyperbranched structure. The cement particles had stronger attractive interaction and formed large flocs at the lower w/c ratio. For hyperbranched PCEs, which possess a dense and rigid conformation in solution, the conformation adjusts less when adsorbed; there may be a thick adsorption layer and a reduced particle attractive interaction, which prevented them from forming large flocs. It can be inferred that the hyperbranched PCEs had a stronger adsorption and dispersive driving force than the comb-like PCE at the lower w/c ratio, which is apparently caused by the flexible conformation of comb-like PCE being more likely to huddle up, and also may be the adsorption layer thickness playing a crucial role in dispersing ability.

### 3.4. Marsh Time of Cement Paste

As seen in Figure 5, the Marsh time increased with the increase of w/c, and hyperbranched PCEs could reduce the apparent viscosity of cement paste. The Marsh time of cement paste containing PCEs followed the order of PM-3 < PB-1 < PM-2 < PM-4 < PM-1 < PD-3 < PD-2 < PD-1 < P2 < P1 at both w/c, implying that the apparent viscosity of cement paste with hyperbranched PCEs was lower than that with comb-like PCE. The Marsh time of cement paste with PM series PCEs decreased with the increased phosphate diester. That is to say, the higher the degree of PCE branching, the shorter the Marsh time and the more excellently reduced the viscosity of the cement paste is. However, PM-4 had a higher branching degree than PM-2, but PM-4 had a longer Marsh time due to PM-4 having stronger hydrophobicity than others. It was prone to intermolecular and intramolecular hydrophobic associations to form mixed aggregates, which may increase the viscosity of cement paste. The results of the phosphate monoester on the effect of Marsh time indicated that the Marsh time of the cement paste increased with the increase of phosphate monoester except PM-1. In other words, the Marsh time decreased with the degree of increase of the PCE branching. PM-4 had a shorter Marsh time than the PD series, due to the fact that PM-1 had a higher charge density and thicker adsorption layer thickness, suggesting that the higher the branching degree of PCE, the stronger the reducing viscosity ability. When the PCE had a lower degree of branching, it had a higher charge density, and it had a stronger reducing viscosity ability.

### 3.5. Viscosity of Aqueous Solution and Simulated Pore Solution

The viscosity of a cement suspension could be described by the Krieger–Dougherty Equation [36,37].

Figure 6 presents the viscosity of aqueous and simulated pore solutions containing PCEs. The viscosity of PCEs’ solutions decreased with the degree of increase of the PCE branching. Therefore, even if the hyperbranched PCEs had a higher residual concentration in pore solutions than comb-like PCE, the viscosity of solutions was still lower than that of comb-like PCE due to the lower intrinsic viscosity of hyperbranched PCEs. This is also because the inorganic ions from pore solutions affect the conformation of PCEs. This result could also explain why hyperbranched PCEs possess a viscosity-reducing performance.

### 3.6. Rheology of Cement Pastes

Shear stress and shear viscosity versus shear rate variations in cement pastes containing PCEs were shown in Figure 7. The shear stress versus shear rate curves shows that the shear stress remarkably increased with an increased shear rate and decreased w/c ratio. Overall, the shear stress decreased while the branching degree of hyperbranched PCEs increased at the same shear rate. Shear stresses of PM-3 and PB-1 were lower than others at the same shear rate, revealing that the lowest viscosity of cement paste is compatible with the Marsh time of the cement paste and viscosity of simulated pore solution. It is interesting that the shear stress of PB-1 is greater than PM-3 at the same shear rate, mainly because the anchor group of PB-1 is less than PM-3, which weakens the repulsive effect between cement particles. On the other hand, the shear viscosity versus shear rate curve exhibited that the shear viscosity decreased while the branching degree of hyperbranched PCEs increased at the same shear rate. The shear viscosity of PM-3 and PB-1 were lower than others, revealing the lowest viscosity of cement paste containing PM-3 and PB-1. However, it is observed that shear viscosity curve of PM-3 and PB-1 can be basically coincided at w/c = 0.29. The shear viscosity curve of PM-3 was below PB-1 at w/c = 0.20, which indicates that the introduction of the hyperbranched structure had an excellent viscosity-reducing performance. The more strongly adsorbed anchoring groups that are introduced, the better the viscosity-reducing performance at the lower w/c ratio. What also deserves to be mentioned specially is that the presence of PCEs bring about a marked shear-thickening behavior, especially at w/c = 0.29, but the first is shear-thinning, then shear-thickening behavior at lower w/c = 0.20. 

### 3.7. Water Film Thickness and Adsorption Layer Thickness

Figure 8 presents the effect of w/c ratio on the water film thickness. Cleary, the water film thickness decreased with the decrease of the w/c ratio due to the decrease of space between particles when the w/c ratio was reduced. The lower space between particles will amplify the van der Waals forces among particles, compressing the water film thickness. It is interesting that the WFT was determined by the w/c ratio and had no relation with the type of PCE.

Figure 9 presents the adsorption layer thickness of PCEs on the nano silica surface. Apparently, the thickness of the adsorption layers of the hyperbranched PCEs was greater than that of the comb-like PCEs, and the adsorption layer thickness increased when the branching degree of PCEs increased. However, PM-4 had the thickest adsorption layer thickness, which may be explained by the fact that the adsorption of PM-4 on the particle surface with residual PM-4 effects was prone to intermolecular hydrophobic associations to form a mixed aggregate. The adsorption layer thickness of PB-1 without a phosphate anchoring group was also thicker due to the high degree of branching and the intermolecular and intramolecular hydrogen bond formed by the introduced N and O. For the effect of phosphate monoester on the adsorption layer thickness, it indicated that the more the number of strongly adsorbed anchoring groups, the greater the adsorption layer thickness of PCE. It was interesting that the thicker adsorption layer thickness of PCEs had an excellent viscosity-reducing performance. However, although PM-4 had the thickest adsorption layer thickness, the viscosity-reducing performance of PM-4 was worse than that of PM-3, which may be because the intrinsic viscosity of PM-4 was too high and there were too many intermolecular hydrophobic associations. 

To further verify the adsorption layer thickness on the particles’ surfaces, the morphology of the adsorption layer, revealed by TEM observation (see Figure 10), indicated that the thickness of the adsorption layers of hyperbranched PCEs was greater than that of comb-like PCEs. The adsorption layer of PM-3 was the thickest due to the high branching degree of PM-3 and the strong adsorption anchoring groups. Meanwhile, the mean adsorption layer thickness of PCEs from DLS was larger than that from TEM due to the hydrated layer formed on silica-CSH with PCE.

### 3.8. Adsorption Behavior

The adsorption amount of PCEs on the cement surface was measured, and the results are listed in Table 4. The results clearly show that the adsorption amount increased while the degree of branching increased. Compared with PM-3, the lower degree of branching of PM-4 had the same adsorption amount, mainly because PM-4 is more prone to hydrophobic association effect. This may also explain why the adsorption layer thickness of PM-4 was thicker than PM-3. Furthermore, the effect of phosphate monoester on the adsorption amount indicates that the adsorption amount of PCEs had no relation with the phosphate monoester density, but the PD-3 had a higher adsorption amount, possibly due to hydrophobic associations. The adsorption amount of PB-1 without the phosphate anchoring group was close to PM-3 owing to the higher degree of branching of PB-1 and the hydrogen bonding effect. This may also explain why the adsorption layer thickness of PB-1 was close to PM-4. As can be clearly seen, the hydrogen bonding and hydrophobic associations can improve the adsorption amount of PCE on the cement particles. Further, the adsorption amounts of hyperbranched PCEs were much higher than those of comb-like PCEs since the hyperbranched structure leads to a dense and rigid conformation in solution, which is less flexible compared to comb-like PCE, thus creating a thicker adsorption layer. Additionally, the more the adsorption amounts of PCEs, the thicker the adsorption layer thickness of PCEs, which indicates that a thicker adsorption layer leads to more free water and a stronger repulsion between particles at a lower w/c ratio. However, with an increased w/c ratio, the water film thickness increased, but the proportion of the adsorption layer thickness to the water film thickness decreased. This may also explain why the dispersing performance of thicker adsorption layer thickness of PCEs was lower at w/c = 0.29.

### 3.9. Viscosity-Reducing Mechanism

It has been widely accepted that the flowability and rheology of cement paste is mainly governed by the water film on cement surfaces and the viscosity of pore solutions. The adsorption layer included the adsorption layer of PCE and the adsorption layer of water. Water exceeding the water film was identified as “free water”. The adsorption of free water onto cement particles lubricated the particles. The deductive mechanism of viscosity reduction is illustrated in Figure 11.

The thicker the adsorption layer thickness, the thinner the diffusion water layer thickness. A thinner water diffusion layer thickness increased the amount of free water and improved the flowability and reduced the viscosity of cement paste. The hyperbranched PCEs led to a smaller occupying area and a higher amount to obtain the required paste spread, the hyperbranched PCEs had a thicker adsorption layer thickness, and the diffusion water layer thickness was compressed. Meanwhile, the hyperbranched PCEs were imbedded with stronger anchoring groups of hydrophobic phosphate ester structure, significantly decreasing the water content of adsorption layer thickness and resulting in the release of more free water, which indicated that the lower viscosity was obtained. The intrinsic viscosity of hyperbranched PCEs and the viscosity of aqueous solutions containing the hyperbranched were lower than that of comb-like PCE, thus lowering the entanglement effect of the side groups or chains on the polycarboxylate-based polymer backbone and decreasing the viscosity of pore solutions in the cement paste.

## 4. Conclusions

The stronger anchoring groups of phosphate monoester and the micro-crosslinked structure formed by phosphate diester in PCE molecules significantly influence the adsorption behavior, dispersing performance as well as viscosity. 

The hyperbranched PCEs had a thicker adsorption layer thickness, lower intrinsic viscosity, and a lower viscosity of simulated pore solution. The thicker adsorption layer thickness raised the cement particle dispersion distance. The low viscosity characteristic of hyperbranched PCEs reduces polymer chain entanglement and viscosity of pore solution. 

## Figures and Tables

**Figure 1 materials-17-01896-f001:**
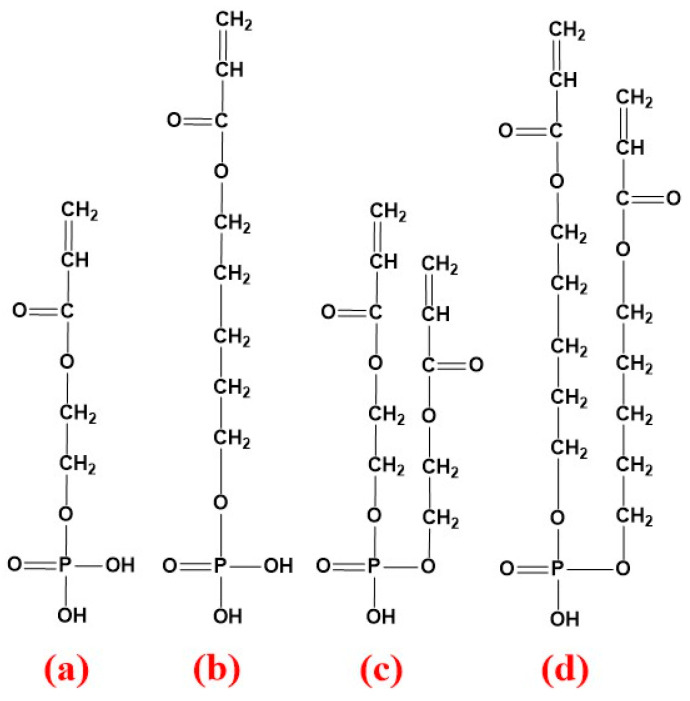
Chemical structure of unsaturated phosphate monomers adopted in synthesis of PCEs: (**a**,**b**) the phosphate monoester of MOEP and HPEP; (**c**,**d**) the phosphate diester of MOEP and HPEP.

**Figure 2 materials-17-01896-f002:**
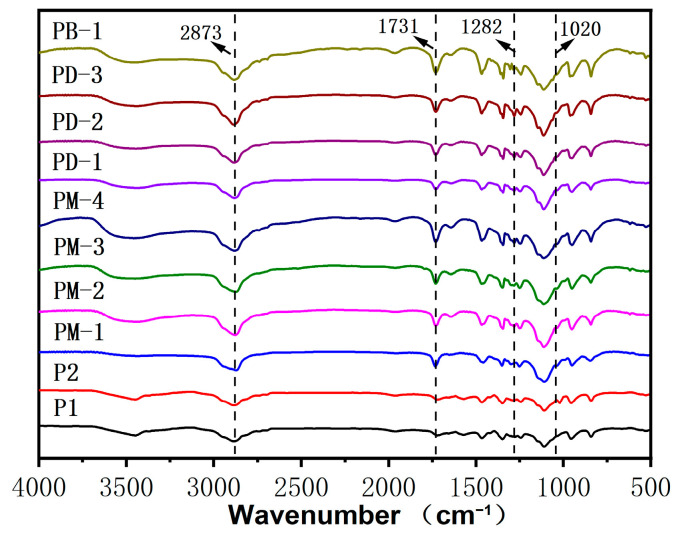
IR spectra of the PCEs.

**Figure 3 materials-17-01896-f003:**
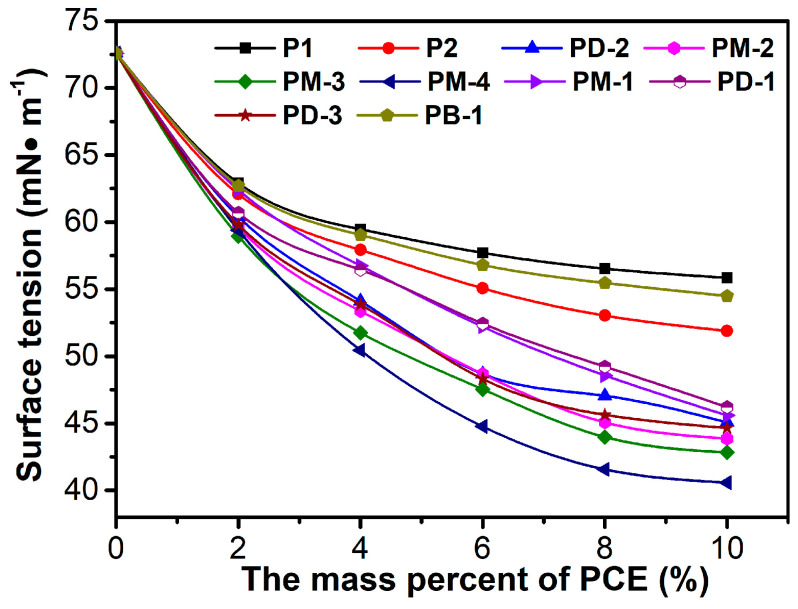
Surface tension of PCEs solution as a function of PCE mass percent.

**Figure 4 materials-17-01896-f004:**
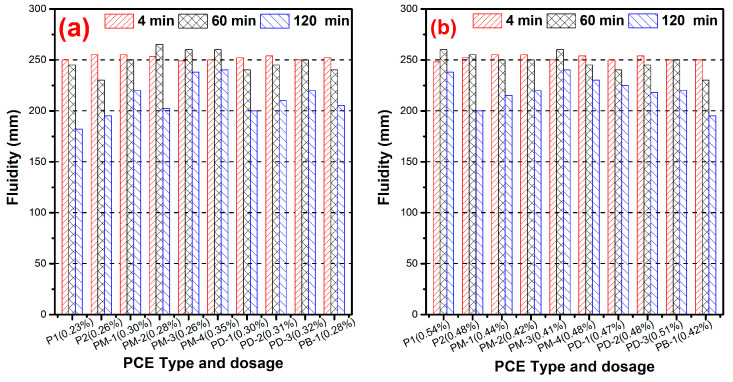
Dispersing ability of PCEs at w/c = 0.29 (**a**) and w/c = 0.20 (**b**).

**Figure 5 materials-17-01896-f005:**
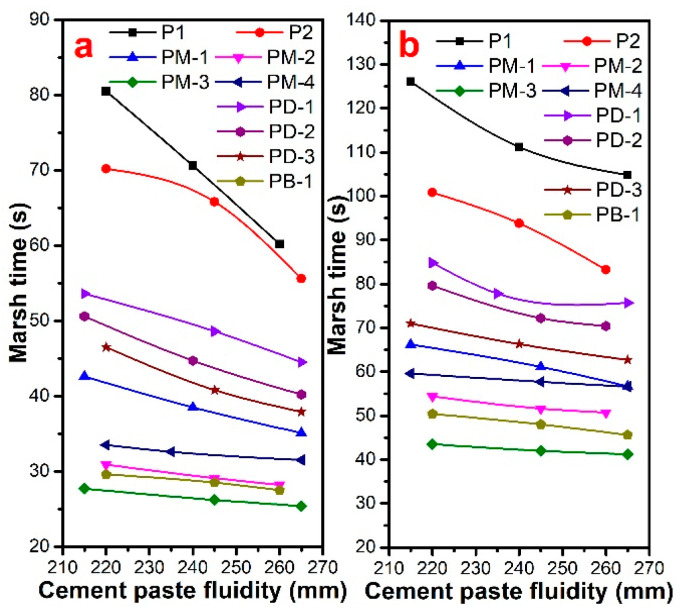
Marsh time of cement paste with different fluidity for PCEs at w/c = 0.29 (**a**) and w/c = 0.20 (**b**).

**Figure 6 materials-17-01896-f006:**
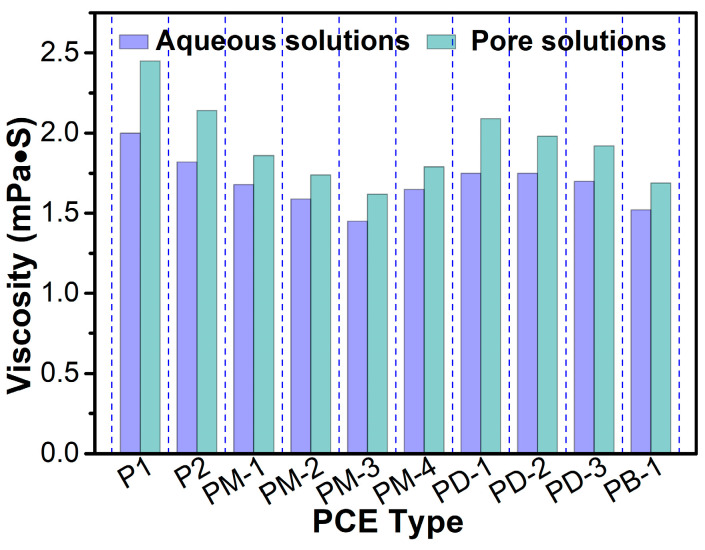
Effect of PCEs type on the viscosity of the aqueous (pH was 6.5) and simulated pore solutions (pH was 12.2).

**Figure 7 materials-17-01896-f007:**
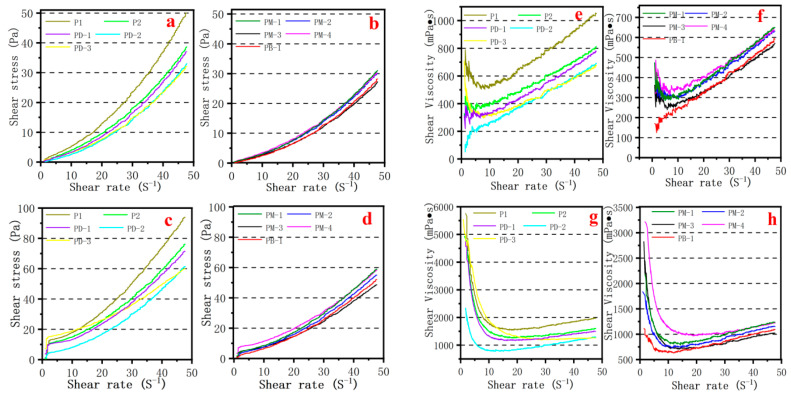
Effect of PCEs type on rheology of cement paste: (**a**,**b**,**e**,**f**) at w/c = 0.29; (**c**,**d**,**g**,**h**) at w/c = 0.20.

**Figure 8 materials-17-01896-f008:**
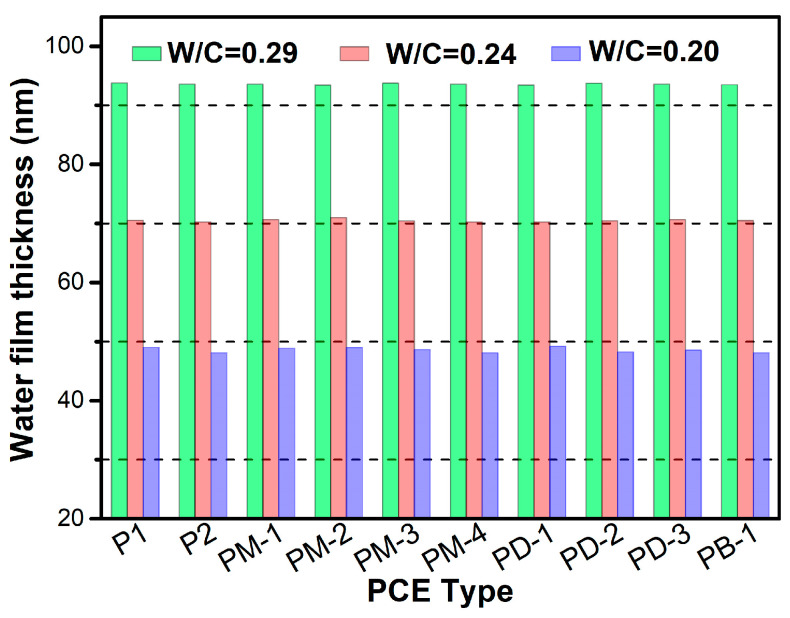
The effect of w/c ratio on the water film thickness.

**Figure 9 materials-17-01896-f009:**
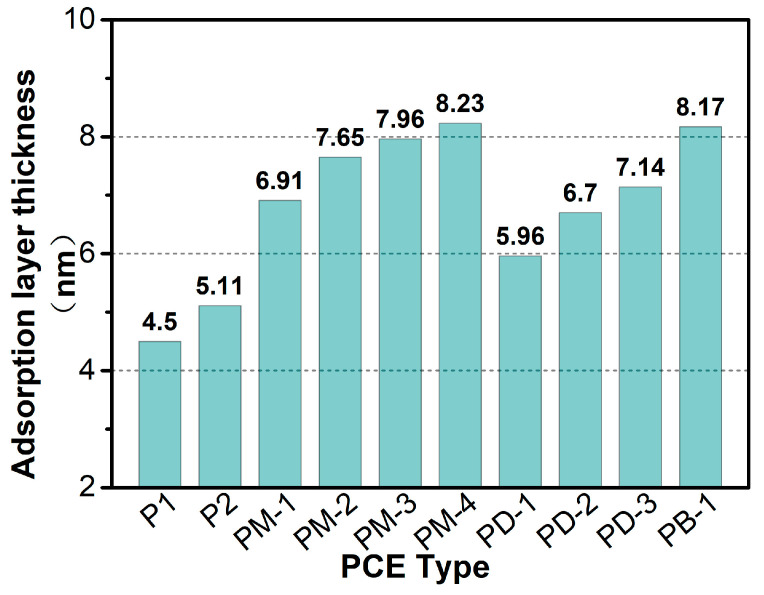
Adsorption layer thickness of PCEs on the nano silica by DLS method.

**Figure 10 materials-17-01896-f010:**
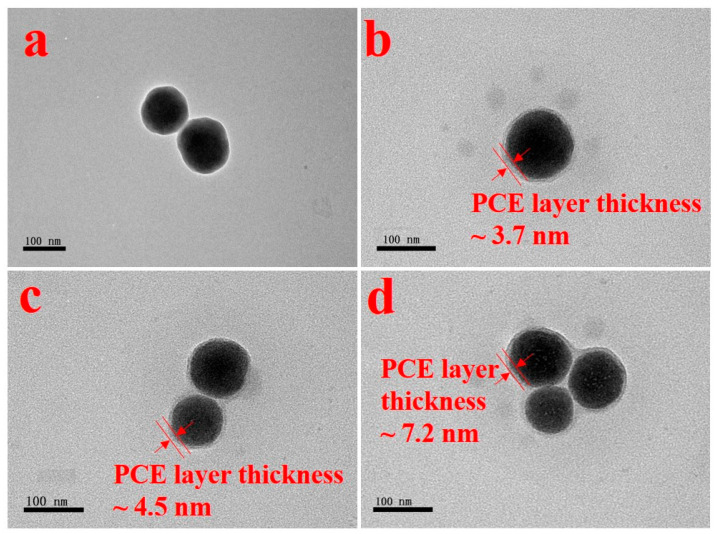
TEM images of silica-CSH (**a**), silica-CSH with P1 (**b**), silica-CSH with P2 (**c**), and silica-CSH with PM-3 (**d**).

**Figure 11 materials-17-01896-f011:**
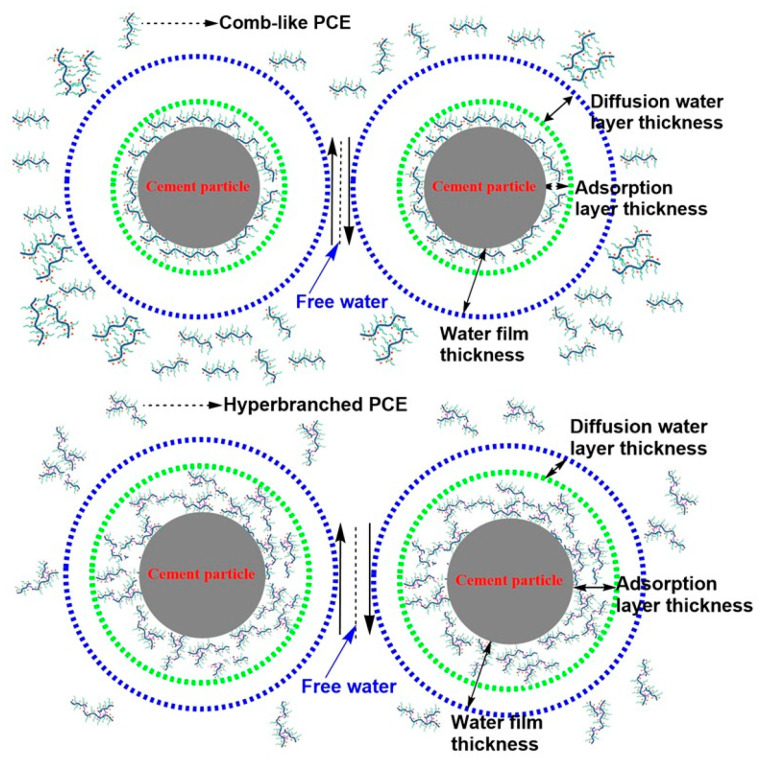
Schematic diagram of mechanism of viscosity-reducing type of PCE.

**Table 1 materials-17-01896-t001:** The composition of the unsaturated phosphate ester samples.

Samples	Phosphate Monoester/%	Phosphate Diester/%	Free Phosphoric Acid/%	Acid Value 1	Acid Value 2
MOEP-1	37.09	59.87	3.04	208.07	76.14
MOEP-2	46.18	49.53	4.30	181.54	91.60
MOEP-3	43.88	37.92	18.19	272.43	169.12
HPEP	33.33	63.94	2.74	90.75	32.73

**Table 2 materials-17-01896-t002:** Chemical composition of cement.

Al_2_O_3_ (%)	CaO (%)	Fe_2_O_3_ (%)	K_2_O (%)	MgO (%)	Na_2_O (%)	SO_3_ (%)	SiO_2_ (%)	TiO_2_ (%)
4.94	63.05	2.92	0.66	1.33	0.15	3.83	19.95	0.27

**Table 3 materials-17-01896-t003:** Monomer composition and structural characterization parameters of PCEs.

PCE Type	Phosphate Ester (g)				MBA (g)	TGA (g)	Mn (10^4^ g/mol)	Mw (10^4^ g/mol)	PDI	[η] (dL/g)	Mark–Houwink α	T_c_ (°C)	T_m_ (°C)
	MOEP-1	MOEP-2	MOEP-3	HPEP									
P1	-	-	-	-	-	0.36	3.42	5.41	1.58	0.31	0.52	18.6	43.8
P2	-	-	-	-	-	0.40	3.35	5.36	1.60	0.26	0.44	22.1	45.5
PM-1	3.62	-	-	-	-	0.48	3.38	5.58	1.65	0.25	0.40	23.2	44.8
PM-2	-	3.44	-	-	-	0.50	3.53	6.99	1.98	0.18	0.34	23.9	46.4
PM-3	-	-	4.29	-	-	0.55	3.68	7.80	2.12	0.16	0.30	26.4	46.9
PM-4	-	-	-	5.72	-	0.46	3.46	6.37	1.84	0.19	0.33	26.4	45.7
PD-1	-	2.78	-	-	-	0.46	3.45	5.69	1.65	0.25	0.42	26.4	49.2
PD-2	-	-	2.30	-	-	0.42	3.41	6.34	1.86	0.24	0.38	25.9	50.2
PD-3	-	-	-	2.71	-	0.42	3.65	6.02	1.65	0.22	0.36	25.8	48.2
PB-1	-	-	-	-	1.23	0.48	3.69	7.05	1.91	0.17	0.32	27.1	50.8

Note: The mole ratios of phosphate monoester and phosphate diester in PM series were 1:0.56, 1:0.70, 1:1.05, and 1:1.19, respectively. The mole ratios of phosphate diester and phosphate monoester in PM-1 and PD series were 1:1.77, 1:1.43, 1:0.95, and 1:0.84, respectively. Mn is the number-average molar mass; Mw is the weight-average molar mass; PDI is polydispersity index; [η] is intrinsic viscosity.

**Table 4 materials-17-01896-t004:** Adsorption amount of hyperbranched and comb-like PCEs.

PCE Type	Concentration before Adsorption		Concentration after Adsorption		Adsorption Amount (mg/g)		Adsorption Efficiency	
	w/c = 0.29	w/c = 0.20	w/c = 0.29	w/c = 0.20	w/c = 0.29	w/c = 0.20	w/c = 0.29	w/c = 0.20
P1	0.23%	0.54%	0.16%	0.46%	0.28	0.32	30.4%	14.8%
P2	0.26%	0.48%	0.18%	0.34%	0.32	0.40	30.8%	20.8%
PM-1	0.30%	0.44%	0.20%	0.33%	0.40	0.44	33.3%	25.0%
PM-2	0.28%	0.42%	0.15%	0.28%	0.52	0.56	46.4%	33.3%
PM-3	0.26%	0.41%	0.08%	0.21%	0.72	0.80	69.2%	48.8%
PM-4	0.35%	0.48%	0.17%	0.29%	0.72	0.76	51.4%	39.6%
PD-1	0.30%	0.42%	0.20%	0.30%	0.40	0.48	33.3%	28.6%
PD-2	0.31%	0.47%	0.19%	0.34%	0.48	0.52	38.7%	27.7%
PD-3	0.32%	0.48%	0.17%	0.32%	0.60	0.64	46.9%	33.3%
PB-1	0.28%	0.51%	0.09%	0.30%	0.76	0.84	67.9%	41.2%

## Data Availability

Data are contained within the article.

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
