# Peer review of "Effects and Mechanism of Hyperbranched Phosphate Polycarboxylate Superplasticizers on Reducing Viscosity of Cement Paste"

_materials, 2024, doi:10.3390/ma17081896_

Round 1
Reviewer 1 Report
Comments and Suggestions for Authors
This is a very interesting research paper. I find it very complete in terms of methodology and discussions. I really appreciated the proposed mechanism trying to explain the viscosity reduction of pastes. My main concerns are about English, which sometimes changes from outstanding to unsatisfactory. I state some recommendations and comments as well:
1. Introduction: in page 2, line 62, authors state: "... and the rheological properties of fresh paste also investigated". I suggest to change "rheological properties" by viscosity, since it was the only rheological parameter measured. For instance, no yield points of pastes or oscillatory measurements were performed.
2. Materials and Methods: In materials section, chemical compounds are recommended to be written using subscript for numbers. In Table 3, even if PDI and [h] are polidispersity index and intrinsic viscosity, a legend is necessary to precise this. In this table (Table 3), the mole ratios between phosphate diester and phosphate monoester are described at the bottom of the table. I recommend describing this information in an additional table, since it is fundamental to understand results.
In the "dispersion of PCEs" section, Figure 4 is cited. However, Figures 2 and 3 are not cited before. Cite figures in order.
Section 2.7: I cannot distinguish the differences between the aqueous and pore solutions. Can authors describe how they were obtained?. In other, the geometry used to determine viscosity using rheometer must be specified. Is the same geometry used to characterize pastes?
3. Results: It is recommended the use of subscripts and superscripts to write compounds and describe wavelengths units in section 3.1. Use CH3 instead of CH3; use cm-1 or 1/cm instead of cm-1.
Section 3.5 "viscosity of aqueous solution and simulated pore solutions". Viscosities are very low, suggesting a Newtonian behavior. However, this has to be commented or the shear rate must be specified. Since all values are very close, errors bars will be very helpful. The geometry is also a required information, since probably applied torque is close to minimal necessary values to obtain good reproducibility.
Section 3.6. I recommend to change "Rheological properties" by "viscosity curves", since this was the only rheological parameter measured in this section. Here again, the geometry is a fundamental parameter that must be provided. I find Figures 7a to 7c useless, since Figures 7e to 7h describe the more important parameter: viscosity. I mean, what is described and discussed in shear-stress against shear-rate figures, is best noticed when using viscosity curves. In all cases, a log-log scale is necessary to improve the visualization of the evolution of viscosity with shear rate. Authors will observe a shear thinning and then a shear thickening behavior, and discussion can be also improved.
Figure 9: I guess units for adsorption layer thickness (y-axis) are also "nm". This must be specified.
Section 3.9. In page 13, line 402, authors affirm: "... and lubricate the particles". Lubrication is usually due to the adsorption of molecules at the solid/liquid interface (i.e. the adsorption layer). Please, revise this.

Comments on the Quality of English LanguageEnglish must be enhanced. There are many spelling and editing errors. Some of them are pointed out in the attached document.
Author Response
Dear Professors,
Thank you very much for your attention to our paper. You are acknowledged for the careful reviews on our manuscript entitled ‘Effects and mechanism of hyperbranched phosphated poly-2carboxylate superplasticizers on reducing viscosity of cement 3paste’ (Manuscript ID: materials-2747790). We really appreciate for your considerate directions and suggestions. These comments are all valuable and very helpful for revising and improving our paper, as well as the important guiding significance to our research. Following your detailed suggestions, we have made a careful revision on the original manuscript. We tried our best to improve the manuscript and made a series of changes in the manuscript. Revised portions are marked in color in the revised manuscript. Below you will find our point-by-point responses to the comments or questions.
Once again, thank you very much for your comments and suggestions.
----------------------------------------------------------------------------------
Report of the First Reviewer
This is a very interesting research paper. I find it very complete in terms of methodology and discussions. I really appreciated the proposed mechanism trying to explain the viscosity reduction of pastes. My main concerns are about English, which sometimes changes from outstanding to unsatisfactory. I state some recommendations and comments as well:
- Introduction: in page 2, line 62, authors state: "... and the rheological properties of fresh paste also investigated". I suggest to change "rheological properties" by viscosity, since it was the only rheological parameter measured. For instance, no yield points of pastes or oscillatory measurements were performed.
Modification description:Thanks for the reviewer’s detailed directions.
We have changed “rheological properties” to be “viscosity”.
Thanks again for your kind directions.
- Materials and Methods: In materials section, chemical compounds are recommended to be written using subscript for numbers. In Table 3, even if PDI and [h] are polidispersity index and intrinsic viscosity, a legend is necessary to precise this. In this table (Table 3), the mole ratios between phosphate diester and phosphate monoester are described at the bottom of the table. I recommend describing this information in an additional table, since it is fundamental to understand results.
Modification description:Thanks for the reviewer’s detailed directions.
We have numbered the chemical compounds in material section.
We have added the legend below Table 3. For example,PDI is polidispersity index; [h] is intrinsic viscosity.
Thanks again for your kind directions.
In the "dispersion of PCEs" section, Figure 4 is cited. However, Figures 2 and 3 are not cited before. Cite figures in order.
Modification description:Thanks for the reviewer’s detailed directions. We have cited Figure 2 and Figure 3 in the revised manuscript, as shown below.
“Fig. 2 shows the FT-IR spectra of comb-like and hyperbranched PCEs.”
“The surface tension of PCEs solution is shown in Fig. 3.”
Section 2.7: I cannot distinguish the differences between the aqueous and pore solutions. Can authors describe how they were obtained?. In other, the geometry used to determine viscosity using rheometer must be specified. Is the same geometry used to characterize pastes?
Modification description:Thanks for the reviewer’s detailed directions.
We have added content of Section 2.7, as shown below.
“The viscosity of the aqueous solution and pore solution was also tested using a Physica MCR 302 (Anton Paar, Austria) rotational rheometer. The viscosity of aqueous solution and simulated pore solution was measured ten times to obtain the average value and standard deviation. However, the viscosities of aqueous solution and simulated pore solution were measured with the rotational rheometer at the concentrations of PCE of 6.7 mg/L and 13.4 mg/L. The concentration was selected based on the viscosity-reducing PCEs accounts for 30% to 60% of PCE.”
- Results: It is recommended the use of subscripts and superscripts to write compounds and describe wavelengths units in section 3.1. Use CH3instead of CH3; use cm-1or 1/cm instead of cm-1.
Modification description:Thanks for the reviewer’s detailed directions.
We have rewritten the compounds to describe wavelengths units, as shown in the revised manuscript.
We have Used CH3 instead of CH3; used cm-1 instead of cm-1, as shown in the revised manuscript.
Section 3.5 "viscosity of aqueous solution and simulated pore solutions". Viscosities are very low, suggesting a Newtonian behavior. However, this has to be commented or the shear rate must be specified. Since all values are very close, errors bars will be very helpful. The geometry is also a required information, since probably applied torque is close to minimal necessary values to obtain good reproducibility.
Modification description:Thanks for the reviewer’s detailed directions.
Section 3.6. I recommend to change "Rheological properties" by "viscosity curves", since this was the only rheological parameter measured in this section. Here again, the geometry is a fundamental parameter that must be provided. I find Figures 7a to 7c useless, since Figures 7e to 7h describe the more important parameter: viscosity. I mean, what is described and discussed in shear-stress against shear-rate figures, is best noticed when using viscosity curves. In all cases, a log-log scale is necessary to improve the visualization of the evolution of viscosity with shear rate. Authors will observe a shear thinning and then a shear thickening behavior, and discussion can be also improved.
Modification description:Thanks for the reviewer’s detailed directions.
We have changed "Rheological properties" to "viscosity curves" in the revised manuscript.
Figure 9: I guess units for adsorption layer thickness (y-axis) are also "nm". This must be specified.
Modification description:Thanks for the reviewer’s detailed directions.
We have added the units “nm” in the y-axis of Fig.9.
Section 3.9. In page 13, line 402, authors affirm: "... and lubricate the particles". Lubrication is usually due to the adsorption of molecules at the solid/liquid interface (i.e. the adsorption layer). Please, revise this.
Modification description:Thanks for the reviewer’s detailed directions.
We have revised it, as below.
“The adsorption of free water onto cement particles lubricated the particles.”
Reviewer 2 Report
Comments and Suggestions for Authors
The authors of manuscript ‘Effects and mechanism of hyperbranched phosphated polycarboxylate superplasticizers on reducing viscosity of cement paste’ studied the polycarboxylate superplasticizers containing phosphate and its effects on the rheological properties of cement paste. This manuscript requires following major changes.
1. Title needs to be revised: ‘phosphated’ is not a word in English. It is highly recommended to change ‘phosphated polycarboxylate superplasticizers’ in title.
2. Introduction. Authors should give a brief introduction about synthesis and chemistry of polycarboxylate superplasticizers.
3. Provide the full form of all abbreviations used in Table 3.
4. It is mentioned that the rheological properties of fresh cement paste was studied. In this case author should provide details of how the cement paste was synthesize including the environmental parameters, such as temperature and humidity at which the paste was mixed.
5. In section 3.1, FTIR discussion: each peak discussed in text should be marked in the IR spectra for the ease of readers.
6. In section 3.1, FTIR discussion: authors identified groups in IR spectra without any reference. It is hard to validate.
7. What’s lacking in this paper is, authors did not discuss other published studies. The results section needs to be described scientifically and author should give a comparison on the findings in this study and related literatures.
8. Additionally, the explanation provided to support their finding is not linked to any study, if it is, then authors should cite the reference study.
9. This paper discussed the effects of PCE on the rheological properties of cement which can be the limitation of this study, but concluding this study without knowing its effects on the hardened properties of cement is not appropriate. Kindly provide recommendation for future study.
1. Make sure your conclusions section restate the topic and its significance, also, the conclusion should underscore the applicability and future study.
1. Lastly, authors should revise the script to ensure the spellings are correct.
Comments on the Quality of English LanguageOverall the readability is okay but script should be revised to ensure the grammar and spellings are correct.
Author Response
Dear Professors,
Thank you very much for your attention to our paper. You are acknowledged for the careful reviews on our manuscript entitled ‘Effects and mechanism of hyperbranched phosphated poly-2carboxylate superplasticizers on reducing viscosity of cement 3paste’ (Manuscript ID: materials-2747790). We really appreciate for your considerate directions and suggestions. These comments are all valuable and very helpful for revising and improving our paper, as well as the important guiding significance to our research. Following your detailed suggestions, we have made a careful revision on the original manuscript. We tried our best to improve the manuscript and made a series of changes in the manuscript. Revised portions are marked in color in the revised manuscript. Below you will find our point-by-point responses to the comments or questions.
Once again, thank you very much for your comments and suggestions.
The authors of manuscript ‘Effects and mechanism of hyperbranched phosphated polycarboxylate superplasticizers on reducing viscosity of cement paste’ studied the polycarboxylate superplasticizers containing phosphate and its effects on the rheological properties of cement paste. This manuscript requires following major changes.
- Title needs to be revised: ‘phosphated’ is not a word in English. It is highly recommended to change ‘phosphated polycarboxylate superplasticizers’ in title.
Modification description:Thanks for the reviewer’s detailed directions.
We have revised the title to be “Effects and mechanism of hyperbranched phosphate polycarboxylate superplasticizers on reducing viscosity of cement paste”
- Authors should give a brief introduction about synthesis and chemistry of polycarboxylate superplasticizers.
Modification description:Thanks for the reviewer’s detailed directions.
We have added more introduction about synthesis and chemistry of polycarboxylate superplasticizers, such as:
“Recently, PCEs have been adopted into concrete in the construction industry because of their great water reduction performances and environmental friendliness in fabricating [1,2]. PCEs have comblike molecular structures and disperses cement grains owing to electrostatic effects combined with steric repulsive forces. When PCE is applied, one part is adsorbed onto mineral surfaces, the second part is integrated into hydrates, and the rest remains in pore solutions. The dispersing property of PCE is found to be significantly associated with its adsorption performance, while there are new findings that nonadsorbing polymers also make a contribution to the dispersing of cement suspension owing to lubrication actions. The confirmation of the adsorbed PCE polymers is highly related to their dosages. The adsorbed PCE immediately contributes to the dispersing forces between cement grains.”
- Provide the full form of all abbreviations used in Table 3.
Modification description:Thanks for the reviewer’s detailed directions. We have added the full form of the abbreviations below Table 3, shown below.
“Mn is the number-average molar mass; Mw is the weight- average molar mass; PDI is polidispersity index; [h] is intrinsic viscosity.”
- It is mentioned that the rheological properties of fresh cement paste was studied. In this case author should provide details of how the cement paste was synthesize including the environmental parameters, such as temperature and humidity at which the paste was mixed.
Modification description:Thanks for the reviewer’s detailed directions. We have added detailed information about the sample preparation and the curing condition and the test process, shown below.
“The conventional comb-like PCE (P1) was synthesized by monomer solution (which consisted of the recipe of P2 without adding DMAEMA), oxidant solution and reductant solution (which consisted of 0.35 g of ascorbic acid and 30.00 g of deionized water). The monomer, oxidant and reductant solution were added dropped into the reaction flask for 2 h, 2.5 h and 2.5 h, respectively. Afterwards, the polymerization was carried out at 50 °C for 2 h. Then the PCE was cooled, neutralized and diluted.”
“2.3 Characterization of PCEs
The PCEs were purified by dialysis via a 3500 Da cellulose membrane. After drying in a vacuum freeze dry box, the PCEs were adopted for further testes.
The FT-IR spectra were measured by a Nicolet Avatar 370 spectrometer. The PCEs were mixed with KBr and pressed into flakes. The molecular weight, PDI, intrinsic viscosity ([η]) and hydrodynamic radius of the synthesized PCEs were determined by GPC using a Viscotek GPCmax system. The hydrodynamic radius was determined by dynamic light scattering (DLS, ALV/CGS-3, Germany). The crystallizing point (Tc) and melting point (Tm) were carried out using a DSC (Q2000, TA Instruments CO., USA). The phase transition temperatures were recorded when the values of the DSC curves were maximized. The surface tension of PCE solution was conducted on an OCA40 Micro surface contact angle tester (Dataphysics Co., Germany), with the measurement range of 0.01~2000 mN/m and measurement accuracy of ±0.01mN/m. The samples are dissolved in diluted water to prepare solution in mass concentration from 0% to 10%.
2.4 Dispersion of PCEs
Cement paste was mixed at W/C of 0.29 at 20℃, the fluidity was determined via mini slump experiment. The Marsh time of cement paste was measured through the funnel. The rheological parameters were measured using a Schleibinger rheometer (Viskomat NT, Schleibinger Company, Germany) at 20 ℃. Water film thickness (WFT), which is defined as the thickness of the water film wrapping the solid particles.
2.5 Adsorption layer thickness
Adsorption layer thickness (ALT) was measured on simulated cement dispersion. The simulated cement dispersion prepared process can be divided into two steps. Firstly, the nano-silica (nano-SiO2) was synthesized by a seeded Stöber method. Secondly, the surface of nano-silica was modified with Ca2+ to simulate the surface of cement hydration products. The detailed preparation process following the reference Huang [11] reported. The particle size of silica-CSH before and after the adsorption of PCEs were measured by dynamic light scattering (DLS, ALV/CGS-3, Germany).
The morphology of adsorption layer thickness was observed through Transmission Electron Microscope (TEM, Tecnai G2 F30 S-Twin, FEI, American) operated at 300kV.
2.6 Apparent Adsorption amount
For measuring the true adsorption of PCEs, adsorption test was performed at w/c=0.29 and w/c=0.20, and the initial fluidity paste was controlled 250 ±5 mm. The fresh cement paste was immediately centrifuged at 3000 rpm for 10 min. Then, the paste was filtered immediately using a microporous filtering film with 0.45 μm. The supernatant solution was diluted with deionized water to suitable concentration for total organic carbon analyzer (TOC, Multi N=C 3100, Jena Company, Jena, Germany) [35].
2.7 Viscosity of aqueous solution and simulated pore solution
The viscosity of the aqueous solution and pore solution was also tested using a Physica MCR 302 (Anton Paar, Austria) rotational rheometer. The viscosity of aqueous solution and simulated pore solution was measured ten times to obtain the average value and standard deviation. However, the viscosities of aqueous solution and simulated pore solution were measured with the rotational rheometer at the concentrations of PCE of 6.7 mg/L and 13.4 mg/L. The concentration was selected based on the viscosity-reducing PCEs accounts for 30% to 60% of PCE.”
- In section 3.1, FTIR discussion: each peak discussed in text should be marked in the IR spectra for the ease of readers.
Modification description:Thanks for the reviewer’s detailed directions. The characteristic peaks of FTIR spectrum have been marked.
- In section 3.1, FTIR discussion: authors identified groups in IR spectra without any reference. It is hard to validate.
Modification description:Thanks for the reviewer’s detailed directions. We have added related references about the identified groups in IR spectra.
- What’s lacking in this paper is, authors did not discuss other published studies. The results section needs to be described scientifically and author should give a comparison on the findings in this study and related literatures.
Modification description:Thanks for the reviewer’s detailed directions. We have added related references about the comparison about the search.
- Additionally, the explanation provided to support their finding is not linked to any study, if it is, then authors should cite the reference study.
Modification description:Thanks for the reviewer’s detailed directions. We have added related references about the comparison about the search.
- This paper discussed the effects of PCE on the rheological properties of cement which can be the limitation of this study, but concluding this study without knowing its effects on the hardened properties of cement is not appropriate. Kindly provide recommendation for future study.
Modification description:Thanks for the reviewer’s detailed directions. We would continue to study effects of the novel PCE on the hydration performance of cement paste, even cementitious materials, in the near future.
- Make sure your conclusions section restate the topic and its significance, also, the conclusion should underscore the applicability and future study.
Modification description:Thanks for the reviewer’s detailed directions. We have rewritten the conclusion part to focus on the key findings and significance of the research.
- Lastly, authors should revise the script to ensure the spellings are correct.
Modification description:Thanks for the reviewer’s detailed directions. We have checked the whole manuscript, and corrected the wrong spelling, and the revised part has been marked in color.
Round 2
Reviewer 2 Report
Comments and Suggestions for Authors
Authors have revised the manuscript, it can be accepted in present form.
Comments on the Quality of English LanguageThe manuscript is readable but the minor Grammar check should be done prior to publication.